# Immune-Onco-Microbiome: A New Revolution for Gynecological Cancers

**DOI:** 10.3390/biomedicines11030782

**Published:** 2023-03-04

**Authors:** Chiara Di Tucci, Ilaria De Vito, Ludovico Muzii

**Affiliations:** 1Department of Maternal and Child Health and Urological Sciences, Sapienza University, 00161 Rome, Italy; 2Independent Researcher, Los Angeles, CA 91101, USA

**Keywords:** microbiome, microbiota, immunity, gynaecological tumours, gynaecological cancers, cancer immunotherapy

## Abstract

Despite significant advances in understanding the pathogenetic mechanisms underlying gynaecological cancers, these cancers still remain widespread. Recent research points to a possible link between microbiota and cancer, and the most recent attention is focusing on the relationship between the microbiome, the immune system, and cancer. The microbiome diversity can affect carcinogenesis and the patient’s immune response, modulating the inflammatory cascade and the severity of adverse events. In this review, we presented the recent evidence regarding microbiome alterations in patients with gynaecological tumours to understand if the link that exists between microbiome, immunity, and cancer can guide the prophylactic, diagnostic, and therapeutic management of gynaecological cancers.

## 1. Introduction

Over 100 trillion microbes inhabit the human body. The term “human microbiota” is the set of symbiotic microorganisms that coexist with the human organism without damaging it. The term “microbiome” refers to the entire microbiota habitat, including microorganisms, their genomes, and the surrounding environment. The microbiome has been identified in the gut, oral cavity, vagina, respiratory tract, skin, and other mucosal surfaces [1].

The microbiome consists of symbiotic microbes that are beneficial to both the human body and microbiota and some, in smaller numbers, of pathogenic microbes that promote diseases. Some of the symbiotic microbes help perform essential functions of the body, such as absorbing nutrients, modulating the immune system, and protecting against pathogenic insults [2].

If a balance alteration of the microbes occurs, for example, during infectious, as a result of certain diets, or after prolonged use of antibiotics, dysbiosis occurs, resulting in the body becoming more susceptible to diseases [3] Figure 1.

The microbiome was officially recognized to exist in the late 1990s. Traditionally, bacteria were studied with microscopy-dependent techniques. In recent years, the advancements in sequencing technology have increased our knowledge of the human microbiome and its implications on health and disease. Several organs previously considered populated by a few bacterial species are now considered to be colonized by approximately 200–300 microbial species [4].

The microbe 16S ribosomal RNA (rRNA) has been characterized with polymerase chain reaction (PCR), DNA hybridization, or fingerprinting. Recently, the advancement of the next-generation sequencing (NGS) 16S method has facilitated the identification of new or lesser-known bacterial species [5].

Through sequencing, it has been possible to characterize most of the population of bacteria living in the human body. The gastrointestinal (GI) microbiome is typically populated by five major bacterial phyla: *Firmicutes, Bacteroidetes, Actinobacteria, Proteobacteria,* and *Fusobacteria*. These bacteria account for approximately 90% of the total microbiota in the gut.

In contrast, a healthy vaginal microbiome is typically populated by members of the *Firmicutes* phylum dominated by *Lactobacillus crispatus*, *Lactobacillus gasseri*, *Lactobacillus iners*, and *Lactobacillus jensenii* [6,7].

Both the relative abundance of the GI and vaginal bacteria may change over time, as well as in different conditions.

A myriad of host-derived factors, depending on the organ site, can influence the balance of the microbiome community. These factors include available nutrients (e.g., diet), hormone levels, host genetics, race, and age. Overall, the microbiome is essential for human development, immunity, and nutrition. The bacteria living with us are mostly beneficial colonizers. In fact, autoimmune diseases such as diabetes, rheumatoid arthritis, muscular dystrophy, multiple sclerosis, and fibromyalgia are associated with microbiome alteration [2].

The human microbiome can also be altered by cancer and cancer treatment. The commensal gut microbiota and their metabolites are involved in oncogenic signalling or may have a suppressive oncogenic function [8].

The aim of the work was to present the recent evidence regarding microbiome alterations in patients with gynaecological tumours to understand if the link exists between microbiome, immunity, and cancer.

Despite significant advances in understanding the pathogenetic mechanisms underlying gynaecological cancers, these cancers still remain widespread, with anticipated 604,000 new cases and 342,000 deaths worldwide in 2020. Cervical cancer is the fourth most frequently diagnosed cancer, with 604,000 new cases and 342,000 deaths worldwide in 2020. Ovarian cancer has recently affected 313,959 women worldwide, resulting in 207,252 deaths; endometrial cancer is the sixth most commonly occurring cancer in women, and there were more than 417,000 new cases of endometrial cancer in 2020 [9].

## 2. Microbiome and Cancer

Recent studies in mouse models provided strong evidence of dysbiotic gut microbiota as a trigger for cancer development, particularly colorectal and liver cancers.

D’asheesh TI et al. described the changes in intestinal microbiota in patients with colorectal cancer (CRC) with a 2.2-fold increase in E. faecalis compared with healthy subjects (*p* = 0.0013) [10].

Lee et al. recently highlighted studies on the effects of meat intake and fermented foods on the characteristics of gut microbiota that can influence colitis-associated factors underlying the progression of CRC [11].

Sobhani et al. reported precancerous lesions and epigenetic changes related to the development of cancer after transplanting faecal material from patients with colorectal cancer into germ-free mice [12].

Different pathological mechanisms have been implicated in bacteria-related carcinogenesis. Inflammation, barrier failure, and dysbiosis can lead to cancer [13]. The microbiota is related to multiple pattern-recognition receptors (PRRs). These PRRs regulate the microbiota as antibacterial mediators as well promoting cell death. The most common PRRs are Toll-Like Receptors (TLRs) which are activated in several cell types, including macrophages, myofibroblasts, epithelial cells, and tumour cells.

Nevertheless, it is the interaction between the microbiome, inflammation, and genes that cause cancer development by inducing DNA damage and changes in metabolite production [13,14]. 

Laniewski et al. proposed four different mechanisms by which bacteria may induce carcinogenesis: by promoting cell proliferation or cellular death, by changing the metabolism within a host cell, and by perturbation of the immune system. *Bacteroides fragilis*, for example, can induce DNA damage through the activation of reactive oxygen species, and it produces proteins involved in cell proliferation [15] (Figure 2).

Microbiota is unique for each organ and tissue, suggesting that microbiota tumour-related mechanisms may be organ-specific [12].

There is also evidence that microbial dysbiosis in the gastrointestinal tract might also induce tumorigenesis in other organs inhabited by microorganisms, such as the skin, oral cavity, and genital tract [16]. 

## 3. Microbiome in Gynaecologic Cancer

Most of the bacteria are in the gastrointestinal tract, but the urogenital tract is also dominated by specific microbiota. The old conception that only the vagina was populated by bacteria has been dismantled by recent discoveries about abundant microbial communities in the uterus, fallopian tubes, and ovaries (Table 1) [17].

Elements contributing to dysbiosis in the gynaecologic tract are genetic factors, lifestyle, and environmental factors [18]. The *Firmicutes *and* Lactobacillus* species are the prevalent bacteria in the vagina microbiome, and a decrease in these commensals and an increase in anaerobic bacteria are associated with bacterial vaginosis and other inflammatory gynaecologic conditions [19]. 

### 3.1. Microbiota and Cervical Cancer (CC)

Recent evidence suggests that the vaginal microbiome is implicated in cervical carcinogenesis [20,21]. It is known that the high-risk HPV genotypes, such as *HPV 16 or HPV 18,* are oncogenic factors in CC. 

Bacterial vaginosis (BV), anaerobic bacteria and non-*Lactobacillus*-dominant vaginal microbiome have been associated with an increased risk of HPV acquisition, persistence, and decreased clearance [22]. Different bacteria species, such as *Gardnerella, Atopobium, Prevotella, Megasphaera, Parvimonas, Peptostreptococcus, Anaerococcus, Sneathia, Shuttleworthia,* are associated with HPV-affected microenvironment, dysplasia or cancer in particular. *Sneathia,* a member of the phylum *Fusobacteria*, was the most important microorganism implicated in cervical carcinogenesis, and its presence can represent a meta-genomic marker for HPV persistence and progression of cervical neoplasm [20].

A recent review reported an association between non-*Lactobacillus*-dominant vaginal microbiome dysbiosis and carcinogenesis by predisposing women to HPV acquisition (overall RR 1.33; RR among young women 1.4), persistence, and consequent precancerous dysplasia (RR 1.14; RR 2.01; I2 0%) [23]. 

Norenhag et al. conducted a network meta-analysis that underlined that women with vaginal microbiome dominated by *Lactobacillus iners* and *gasseri* (OR, 3.3; 95% CI) had two to three times higher odds of high-risk HPV infection and cervical neoplasia than women with a vaginal microbiome dominated by *Lactobacillus crispatus* [24]. These results were confirmed by a subsequent meta-analysis, in which *Lactobacillus crispatus*, but not *Lactobacillus iners*, was related to lower detection of high-risk HPV (OR 0.49; 95% CI; I2 10%) and dysplasia (OR 0.50; 95% CI; I2 0%) [25]. 

After investigating the cervical metagenomes in the case of Cervical Intraepithelial Neoplasia (CIN) and CC, Kwon et al. showed that *Lactobacillus,* Staphylococcus, and *Candidatus endolissoclinum* were prevalent in CIN2/3, while *Alkaliphilus* and *Wolbachia* were the prevalent species in the case of CC [26]. Kang et al. presented a prevalence of *Prevotella* in faecal samples of women with early CC. Recent studies theorize that the gut microbiome can induce the growth of CC through an inflammatory response mediated by the activation of TLRs [27].

Laniewski et al. investigated the association of immune response with the microbiome in the progression to malignancy for cancer cells. Women with cervical cancers but not with dysplasia exhibited increased genital inflammatory scores and elevated specific immune mediators; for example, IL-36γ were significantly associated with cervical cancers [28].

### 3.2. Microbiota and Uterine Cancer

The uterus is not a germ-free organ but is colonized through an ascending mechanism from the vagina, and it is dominated by a greater variety of bacterial species and different strains of *Lactobacillus* [17].

In women undergoing total hysterectomy, after eliminating the vaginal contamination, *Lactobacillus* species were detected in the endometrium, similar to microorganisms of the vagina. In contrast, Winters et al. recently, through 16S rRNA gene qPCR and sequencing, reported that microbiota in the middle endometrium is not dominated by *Lactobacillus* as was previously concluded but is dominated by *Acinetobacter, Pseudomonas, Cloacibacterium, Escherichia, and Comamonadaceae* [29]. A strong correlation has been demonstrated between gut microbiota, estrogen metabolism, and obesity [30].

Estrogens may induce alterations of vaginal microbial communities and play an important role in modulating the inflammatory response with the production of pro-inflammatory molecules, for example, TNF alpha ad IL-6 [31]. Conjugated estrogens excreted in the bile can be modified by bacterial species in the intestine, such as estroboloma, which performs its function through the enzymatic action of beta glucuronidase. The estrogen modulatory effect can induce the development of hyperplasia and endometrial cancer (EC) by interfering with the gut–vaginal microbiome axis. Furthermore, EC is promoted by obesity, diabetes, and metabolic syndrome that may promote changes in the microbiome [32]. 

One of the first studies about the association between microbiome and EC was conducted by Walther-Antonio et al. [33]. Culture-independent models, through 16s rDNA, reported the presence of *Atopobium vaginae*, *Porphyromonas* sp. and a high vaginal pH in women with EC. The same group confirmed the presence of *Porphyromas somerae* as the most important biomarker for EC [34]. Schreurs et al. described an abundance of *Actinobacteria, Firmicutes*, *Proteobacteria,* and *Bacteroides* in obese women with EC compared to non-obese women [35]. 

A recent study conducted by Lu et al. suggested a link between inflammatory cytokines, bacterial flora, and EC. *Micrococcus* species were found to be associated with the alteration of endometrial microbiota and with the production of inflammatory cytokines. In particular, in the group of EC, IL-6, and IL-17 mRNA levels were elevated [36].

Walsh et al. described a correlation between EC, postmenopausal status, and increased microbial diversity of the lower genital tract. Furthermore, they identified that the presence of *Porphyromas somerae* was associated with type II EC risk and described that the dysbiosis was correlated with menopause, obesity, and high vaginal pH to uterine dysbiosis [34]. A recent work conducted by Gressel et al. described the microbiota of postmenopausal undergoing hysterectomy for endometrioid and uterine serous cancer and demonstrated the microbial diversity of anatomic niches in these women compared to controls [37].

### 3.3. Microbiota and Tubal and Ovarian Cancer

A specific microbiome was also postulated for ovarian and fallopian tube tissues, and in women affected by ovarian cancer (OC), specific bacteria *Firmicutes* were identified, such as *Abiotrophia, Bacillus, Enterococcus, Geobacillus* [38,39]. However, studies about the association between microbiomes and the development of OC are few. Similar to EC, chronic infections and inflammation in the genital tract may induce the development of ovarian tumours [39,40]. A study presented an association of *Brucella* with OC, and another study showed the presence of *Chlamydia* in 70% of these cancer tissues, while *Mycoplasma* was detected in 59% of pathologic ovarian samples. *Chlamydia* may induce cancer through DNA damage, inhibition of apoptosis, and predisposition to other infections [41].

The fallopian tube is a precursor for ovarian carcinogenesis, and recent evidence analysed the microbiome of fallopian tubes as a starting point for ovarian cancers. Zhou et al. showed reduced biodiversity and microbiome richness in OC tissues compared to tissues from normal distal fallopian tubes and proposed that the microbiome may influence the tumour microenvironment in OC through the activation of Treg cells [39]. The change of microbial composition with the increase of *Proteobacteria/Firmicutes* might be associated with the initiation and progression of OC and could regulate the local immune environment. Banerjee et al. showed that the microbiome of ovarian tumours is different from ovarian tissue that has never been in the proximity of cancer. They detected an unexpected and robust microbiome, including members of the bacterial, viral, fungal, and parasitic family and suggested an association with the genesis or propagation of cancer. Banerjee et al. also hypothesised that the tumour microenvironment might provide a favourable milieu for these microorganisms to persist [40]. These works suggested an important link between inflammation and microbiome in the genesis of OC. The latest evidence also described changes in the microbiota at the site distant from the tumour tissue. A recent work conducted by Morikawa et al. analysed the cervicovaginal microbiota of OC women and observed a *Lactobacillus*-poor, highly-diversified microbiota in OC premenopausal women compared to healthy subjects. This study demonstrated that cervicovaginal microbiota could be considered a biomarker of OC in premenopausal women. In particular, a correlation between BRCA1/2 mutations, known to increase OC occurrence rate, and cervicovaginal microbiota could be, in future, an intriguing issue [42].

Therapeutic approaches may alter microbiomes and induce OC progression. As for colon cancer, also for OC, there could be a correlation between surgery and changes in the gut microbiota. Ohigashi et al. described an alteration of microbiota after surgery with an increase of *Enterobacteriaceae, Enterococcus, Staphylococcus, and Pseudomonas* [43]. Tong et al. evaluated the effects of surgery and chemotherapy for OC on microbiomes and described a reduction of *Bacteroidetes and Firmicutes* in faecal samples collected after surgery for OC, while an abundance of the same species was detected before chemotherapy. Conversely, *Proteobacteria* species increased after surgery, but a decrease in the same group and an increase in anaerobic bacteria, such as *Bacteroides, Collinsella, and Blautia,* was discovered [44].

Platinum-based chemotherapy following primary debulking surgery is the standard treatment for OC. Platinum may damage the intestinal mucosa and dysbiosis, in particular, a decrease of *Firmicutes* species, which can be related to side effects of chemotherapy such as body weight loss and cardiac dysfunction [45]. Gram-positive bacteria may also contribute to an alteration of response to anti-cancer activities of cisplatin by the production of inflammatory cells producing reactive oxygen species (ROS). Some mechanisms may influence the efficacy of chemotherapy, in particular translocation, which is the process by which the commensal or pathogenic bacteria pass across the gut barrier into the systemic milieu [46]. Therefore, a microbiome reset may contribute to a better response to chemotherapy and a reduction of collateral effects. Hawkins et al. described that [47] an alteration of the intestinal microbiome, which occurs after the administration of broad-spectrum antibiotics, has an impact on the progression of the tumour and on the response to therapy. As a consequence, there is a worsening of survival linked not only to disease progression but also to platinum resistance. In addition, paclitaxel, used in the case of OC, could influence gut microbiota with a decrease in the number and function of beneficial bacteria and an increase in collateral effect. For example, a reduction of *Akkermansia muciniphila* coloniae could be associated with an increase in neuropathic pain [48]. Wang et al. performed studies in mice and showed that in OC-bearing mice, faecal microbiota transplantation (FMT) of OC patients accelerated the progression of the disease. Faecal microbiota supplementation with *Akkermansia* significantly suppressed neoplastic progression in mice [49].

## 4. Cancer Immunotherapy

Immune cells such as T cells are the first level of defence of the human body against cancer. The immune system naturally regulates the cells’ growth, keeping potential cancer development under control. However, cancer cells have ways to circumvent the immune system. For example, cancer cells may:−Hide from the immune system thanks to genetic mutations. −Silence immune cells using targeted receptors on their surface. −Change normal cells around the tumour, interfering with immune system response to the cancer cells.

Cancer immune treatments have changed the landscape of oncology [50]. Immunotherapy aims to activate the body’s immune system and improve tumour-killing ability.

This can be achieved in a couple of ways:−Activating the immune system to better recognize and attack cancer cells. −Provide the immune system with helpful components, such as synthetic immune system proteins.

There are different immunotherapy strategies as indicated in Table 2 [51,52].

Clinical trials for cancer immunotherapy have been increasing in recent years. These trials involve different therapy types with cell therapy, immunomodulators/check-point inhibitors and cancer vaccines. 

In 2006, the FDA approved the first cancer vaccine in human history: a vaccination against CC (Gardasil). Gardasil prevents the infection of human papillomavirus (HPV) 16/18 for more than five years, decreasing CC incidence [53]. Now, in view of the success of COVID-19 mRNA vaccines, cancer mRNA vaccines are of great interest to the scientific community, and mRNA vaccines are being tested for cervical and OC [54].

According to the Cancer Research Institute, currently there are a couple of Immunotherapies FDA approved for CC and OC.

Bevacizumab (Avastin^®^): is a monoclonal antibody that targets the VEGF/VEGFR pathway and inhibits tumour blood vessel growth; it has been approved in combination with chemotherapy for patients with advanced CC and for patients with early-stage or relapsed OC.Pembrolizumab (Keytruda^®^) is a checkpoint inhibitor that targets the Programmed death-ligand 1 (PD-1/PD-L1) pathway; it has been approved for patients with advanced CC with high PD-L1 expression, high microsatellite instability (MSI-H), or high tumour mutational burden (TMB-H); and for patients with advanced OC with high microsatellite instability (MSI-H), DNA mismatch repair deficiency (dMMR), or high tumour mutational burden (TMB-H). 

Some patients treated with immunotherapy have had great responses to these treatments. In a few cases, following immunotherapy, tumours even disappeared in patients with advanced cancers. However, immunotherapy has been linked to severe side effects that can affect almost any organ in the body. Immunotherapy drugs stimulate the immune system to attack tumour cells and can, in some patients, let the immune system lose control and recognize and attack healthy cells as well. 

The Immune-related side effects of immunotherapy highlight a fundamental difference between these drugs and other cancer treatments: conventional treatments such as chemotherapy kills tumour cells directly, whereas immunotherapy does not [55]. 

## 5. Microbiota, Immunity, and Impact on Cancer Treatment

The correlation between the gut microbiota and the immune system has been demonstrated in studies using germ-free (GF) mice that are devoid of detectable microbiota during their lives. Pattern-recognition receptors on innate immune cells recognize bacteria-derived molecules leading to modulation of systemic immunity with induction of T reg cells through stimulatory effects on myelopoiesis and function of dendritic cells (DCs) with the production of transforming growth factor beta, macrophages, and neutrophils. The loss of commensal bacteria can lead to a decrease in T-reg frequency and an increase in T-helper cells with the production of cytokines and chemokines [56]. 

Biomarkers may be predictive of treatment response in gynaecologic cancer such as tumour genomic and proteomic markers, immune response markers, and tumour microenvironment markers. 

Recent evidence suggests that gut microbiota can be considered a tumour marker that may impact cancer treatment response by affecting immune response during and after chemotherapy (CHT). El Alam et al. studied changes in the diversity and composition of the gut microbiome during and after pelvic chemo-radiotherapy (CRT) for gynaecological cancers; 58 women with cervical, vaginal, or vulvar cancer were analysed. The microbiome analysis was conducted using 16Sv4 rRNA gene sequencing before, during treatment and after 12 weeks [57].

Gut microbiome richness and diversity levels continually decreased throughout CRT, with increases in *Proteobacteria* and decreases in *Clostridiales* and increases in *Bacteroides* species after CRT. After 12 weeks of treatment, gut microbiome diversity returned to baseline, but the structure and composition presented alterations. In addition, it is noted that CRT may induce gastrointestinal toxicity that may be increased by microbiota alteration, and CRT-induced dysbiosis increases the susceptibility to CRT-related gastrointestinal toxic effects [58,59,60].

The interactions between the gut microbiota and the host immune system have been reported. Specific bacteria may promote or suppress the activity of immune checkpoint inhibitors (ICI). 

ICIs have been one of the most recent treatments to demonstrate clinical benefits in many cancers, including recent gynaecological cancers [61,62,63]. Whereas standard treatments such as palliative surgery, radiotherapy, and chemotherapy have failed to control the disease in the advanced stages, good results have been obtained with the use of immunotherapy [64,65,66,67]. However, as described above, many changes in the gut microbiota may induce alteration in the number and functions of gut immune cells, resulting in systemic inflammatory responses [68]. In line with this, it is reasonable to think that the effectiveness of immunotherapy can be affected by alterations in microbiota composition. Recent studies demonstrated the association between the gut microbiota and the anti-tumour effects of ICIs. Multiple possible mechanisms underlying the modulation of anti-tumour immunity by the gut microbiota were suggested, such as the activation of IFN-γ pathways, production of IL-12, induction of the Th1 immune response in the tumour-draining lymph nodes through the activation of dendritic cells (DCs), and the maintenance of regulatory T cells [69].

The first few studies about the correlation between microbiota and immunity were conducted on mice, and the tumour growth decreased in specific pathogen-free (SPF) mice compared to that of germ-free (GF) mice. Faecal microbiota transfer (FMT) from responder patients into GF mice resulted in a better response to ICI. 

In 2018, Matson et al. confirmed the role of microbiota on the efficacy of immunotherapy in metastatic melanoma; in particular, *Bifidobacterium* was involved in enhancing the anti-tumour efficacy of anti-PD-1 therapy. The gut microbiome can be considered an essential mediator for therapeutic activities in ICIs and other cytotoxic agents [70]. The abundance of specific operational taxonomic units (OTUs) and enhanced gut microbiota diversity induced the efficacy of anti-PD-L1 therapy. The non-responder group presented an abundance of *Bacteroides thetaiotaomicron*, *Escherichia coli*. Conversely, the responder group presented an abundance of faecali bacterium with prolonged progression-free survival (PFS) [71]. Wang et al. demonstrated a specific correlation between microbiota and immune activation. Faecal microbiota supplementation with *Akkermansia* significantly suppressed neoplastic progression in mice through the production of acetate and subsequent secretion of interferon γ (IFNγ), upregulation of CD8 + T cells, and antiproliferative action.

The use of antibiotics may be considered the principal risk factor for reduced response to immunotherapy because it may induce change in the gut microbiome.

Pinato et al. investigated the role of antibiotic (ATB) therapy administered before (p ATB) or during (c ATB) immunotherapy in patients with non to small cell lung cancer, melanoma, and other tumour types. In the p ATB therapy group, but not c ATB therapy, there was a worse Overall Survival (OS) (2 vs. 26 months for p ATB therapy vs no p ATB therapy, respectively) and a refractory response to ICI therapy (21 of 26 [81%] vs. 66 of 151 [44%], *p* < 0.001). Multivariate analyses confirmed that the *p* ATB therapy (HR, 3.4; 95% CI, 1.9–6.1; *p* < 0.001) and responses to ICI therapy (HR, 8.2; 95% CI, 4.0–16.9; *p* < 0.001) were associated with OS, independent of tumour site, disease burden, and performance status [72]. Routy et al. also described the influence of microbiomes on the efficacy of PD-1 immunotherapy in epithelial tumours and described how the antibiotic administration near the start of immunotherapy was associated with a worsening in OS and PFS [50]. Microbiome compromission may also alter responses to treatment with CTLA-4 blockade. To date, the impact of ATB on ICI response and clinical outcomes is unclear in women with gynaecologic cancer.

Some studies conducted in patients with non-gynaecological cancers demonstrated that antibiotic treatment may be associated with decreased clinical response and worse oncologic outcomes in patients treated with ICIs [50,73]. Spakowicz et al. evaluated the impact of medication use, also antibiotics, in 609 patients treated with ICIs; patients who received treatment, in particular cephalosporin, up to 100 days prior to ICI initiation presented worse OS. The use of antibiotics was associated with the reduction of OS. In a recent meta-analysis of 2363 patients with non-gynaecologic malignancies from 15 studies, patients exposed to ATB before ICI treatment had significantly reduced PFS and OS [74]. 

Chambers et al. in a retrospective cohort study of women with advanced Epithelial Ovarian Cancer (EOC) undergoing platinum chemotherapy studied the effect of antibiotic treatment on responses to ICIs therapy. ATB treatment was associated with decreased PFS and OS. ATB decreased PFS (17.4 vs. 23.1 months, HR 1.50, 95% CI 1.20–1.88, *p* < 0.001) and OS (45.6 vs. 62.4 months, HR 1.63, 95% CI 1.27–2.08, *p* < 0.001) compared to no ATB. Similarly, in multivariable analysis, all ATB and anti-G + ATB significantly worsened PFS (HR 1.31, 95% CI 1.04–1.65, *p* = 0.02), (HR 1.50, 95% CI 1.07–2.10, *p* = 0.02) and OS (HR 1.52, 95% CI 1.18–1.96, *p* = 0.001), (HR 1.83, 95% CI 1.27–2.62, *p* = 0.001), respectively [75].

The same group recently investigated the role of antibiotic treatment in women with recurrent EC, CC, and OC treated with ICIs. They demonstrated that p ATB was associated with decreased ICI response rate (RR). Similarly, the PFS and OS were also decreased in women with EC and CC treated with p ATB compared with women who did not receive ATB [76].

## 6. Future Directions

### 6.1. Diet

Gut microbiomes may be modified by dietary regimes, and active clinical trials are evaluating if increased fibre intake or personalized dietary interventions may impact the microbiome composition [77]. 

Fibre stimulates the intestinal epithelium to induce mucus secretion, and it may be used by anaerobic bacteria to produce short-chain fatty acids (SCFA), which influence the immune system.

Cancer risk may be modified by diet and may influence immunosurveillance. Recent literature data on women with EC and OC show an important role of the ketogenic diet (KD) with an increase in perceived energy and physical function and a decrease in food cravings [78]. Mediterranean diet (MD), being rich in fibre, has the capacity to induce changes in the composition of the intestinal microbiota leading to an increase of *Lactobacillus, Bifidobacterium,* and *Prevotella*, a reduction of *Clostridium,* and a higher faecal SCFA with an increase in gut microbial diversity. A diet rich in fibres may be associated with an increase in the number of beneficial bacteria such as *Ruminococcus, Bifidobacterium*, and *Lactobacillus*. Studies about the role of microbiota in vegan/vegetarian regimens are controversial because a low abundance of *Bifidobacterium* was detected, but no differences in faecal SCFA levels were presented [79].

In gynaecologic cancers implementing dietary interventions can be safe and feasible. Observational studies have shown that fasting during chemotherapy may be a potential adjuvant factor in reducing adverse effects associated with chemotherapy such as fatigue, weakness, and gastrointestinal side effects.

### 6.2. Fecal Microbial Transplantation

Faecal microbial transplantation (FMT) is becoming widespread for the treatment of non-malignant diseases. Investigating FMT in the context of immune checkpoint inhibitors is being conducted with the rationale that FMT may alter immune activation and improve response to checkpoint blockade [80].

### 6.3. Probiotics

One of the most important interventions to restore intestinal microbiota is the supplementation of probiotics.

Probiotics are primarily able to block the inflammatory cascade through various signalling pathways such as the nuclear factor-κB (NF-κβ) pathway, possibly related to alterations in mitogen-activated protein kinases pattern recognition [81].

Consumption of probiotics may increase response to immunotherapy; studies on murine models demonstrated an improved response to ICI. However, only by understanding the metabolic pathways that induce the microbiome response to immunotherapy will it be possible to identify the bacteria that may provide a real treatment benefit.

The dysbiosis and DNA damage may be reduced by the probiotic supplementation through an alteration of the tumour microenvironment’s immune cell composition in women with colon and breast cancer. Probiotics have also been used to reduce side effects of anti-tumour therapy, such as diarrhoea and mucositis. Targeted probiotics interventions in the setting of combined chemoradiotherapy for cervical and EC may help to reduce symptoms of radiation enteritis and cystitis [82].

## 7. Conclusions

Gynaecological cancers are influenced by chronic diseases for example obesity, diabetes, viral infections, and hormonal disfunctions, and these triggers may be modulated by the host microbiota. 

The discovery of the genitourinary microbiome represents a real revolution in the field of gynaecology. In particular, the knowledge of how the microbiota can modulate the immune response, and consequently the inflammatory response in response to immunotherapy in cancer patients, represents the real challenge of the new millennium. 

Randomized, robust, well-designed studies are needed to understand the relationship that exists between microbiota, immunity, and gynaecological cancer. An in-depth knowledge in this field will certainly enable early diagnosis and personalized medicine with increased efficacy of gynaecological cancers.

## Figures and Tables

**Figure 1 biomedicines-11-00782-f001:**
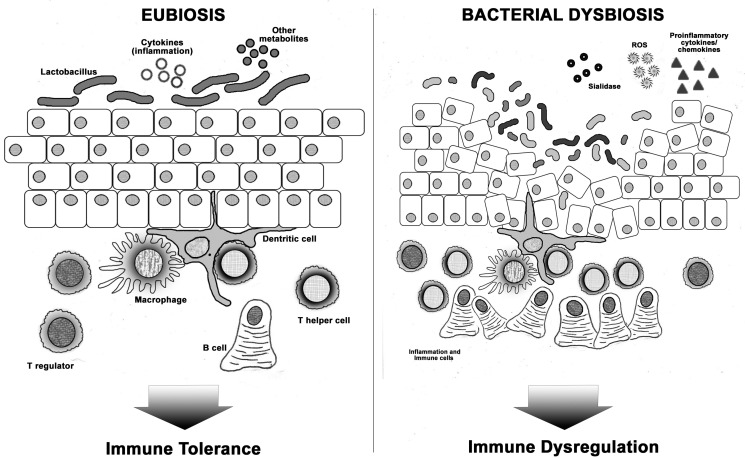
Microbiome and immune system regulation: microbial homeostasis promotes barrier integrity. Bacterial dysbiosis induces mucosal barrier alterations with the release of inflammatory mediators and activation of immune cells.

**Figure 2 biomedicines-11-00782-f002:**
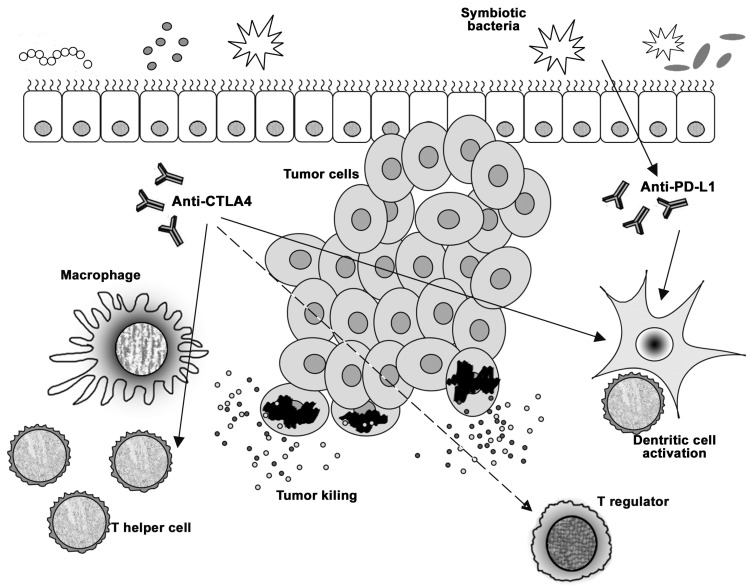
Immune checkpoint inhibitors interact with microbiota and immune cells to enhance anti-tumour activity. Anti-programmed death-ligand 1(PD-L1) treatments, through the interaction with symbiotic bacteria, induce the activation of dendritic cells and promote the activation and function of T-helper cells. Anti-CTLA4 enhances the activity of dendritic cells and T-helper cells while suppressing T regulatory cell function. Solid arrows indicate the activating signal, while dashed arrows indicate the inhibitory signal.

**Table 1 biomedicines-11-00782-t001:** Microrganisms in gynaecologic cancers.

Types of Cancer	Microorganisms
Cervical cancer	*HPV 16, HPV 18, Gardnerella, Atopobium, Prevotella, Megasphaera, Parvimonas, Peptostreptococcus, Anaerococcus, Sneathia, Shuttleworthia, L. iners, L. gasseri, Alkaliphilus, Wolbachia* and *Prevotella*
Uterine cancer	*Acinetobacter, Pseudomonas, Cloacibacterium, Escherichia, Comamonadaceae, Atopobium vaginae, Porphyromonas, Firmicutes, Proteobacteria* and *Bacteroides*
Tubal and Ovarian cancer	*Brucella, Chlamydia, Mycoplasma, Proteobacteria, Firmicutes, Enterobacteriaceae, Enterococcus, Staphylococcus, Pseudomonas, Proteobacteria, Bacteroides, Collinsella* and *Blautia*

**Table 2 biomedicines-11-00782-t002:** Immunotherapies strategies and mechanism of action.

Type of Immune-Therapeutic	Mechanism of Action
Checkpoint inhibitors	Normally, checkpoint proteins, such as Programmed death-ligand 1 (PD-L1) on tumour cells and Programmed Death Protein (PD-1) on T cells, balance the immune response against cancer cells. The binding of PD-L1 to PD-1 prevents T cells from killing cancer cells in the body. By blocking the binding of checkpoint proteins, the T cells can recognize and kill tumour cells.
T cell transfer therapy	This treatment aims to boost the natural ability of the patient’s T cells against cancer. Immune cells are taken from the patient, and the most active are selected or changed in the lab to better attack tumour cells. They are grown in large batches and administered back to the patient through IV
Tumour Vaccines	Vaccine enhance immune system’s response to cancer cells.
Monoclonal antibodies	Recombinant Monoclonal Antibodies are artificially designed to bind to specific targets on cancer cells.
Bispecific T cell engager (BITE) Antibodies	BITE antibodies induce a transient cytolytic synapsis between cytotoxic T cells and the cancer targets.

## Data Availability

Not applicable.

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
