# Peer review of "Immune-Onco-Microbiome: A New Revolution for Gynecological Cancers"

_biomedicines, 2023, doi:10.3390/biomedicines11030782_

Round 1

Reviewer 1 Report

The Authors summarize the current knowledge on microbiome alterations in patients with gynecological cancers and discuss the link between microbiome, immunity, and cancer. In this work, there is also a chapter on future perspectives. In general, the manuscript is well thought out and well organized. The bibliography (most papers from the last decade, up to the 2022 year) is complete. However, there are some points that must be considered.

[Major concern]

- In the Introduction section: State the aim of the work clearly. Some statistics on the incidence of gynecological cancers would be useful here.

- It would be interesting to list (e.g., in the figure or in the table) the most important microorganisms of gynecological cancers, including types of cancer.

[Minor concern]

- No references [3] and [4] on page 2. Instead, there are [44] and [66] (?).

- Lines 47-51: Include an appropriate reference to this paragraph.

- The explanation of the abbreviation for PD-L1 should be on first use – in Figure 2 caption. The same for check-point inhibitors – ICI does not appear until line 328. The explanation for ATB should stay in line 361 (not in line 373).

- ‘Bacteria’ (line 191) and ‘Microbiome’ (line 193) should be lowercase.

- Pay attention to: ‘et al’ or ‘et al.’, placing a reference ‘… [1].’ or ‘… . [2]’ (line 68), ‘microorganisms’ or ‘micro-organisms’ (line 211), ‘checkpoint’ or ‘check-point’, Escherichia Coli (line 353). Please keep the format consistent throughout the whole text.

- There are some typographic errors, mainly a lack of spaces (e.g., lines 16, 69, 132 …). 

Author Response

- In the Introduction section: State the aim of the work clearly. Some statistics on the incidence of gynecological cancers would be useful here: We have addressed this with text addition from line 72 to 81.

- It would be interesting to list (e.g., in the figure or in the table) the most important microorganisms of gynecological cancers, including types of cancer: We have introduced table 1 capturing details about important microorganisms in different type of gynecological cancers, see line 124-125

[Minor concern]

- No references [3] and [4] on page 2. Instead, there are [44] and [66] (?): These have been corrected

- Lines 47-51: Include an appropriate reference to this paragraph: We have introduced the reference.

- The explanation of the abbreviation for PD-L1 should be on first use – in Figure 2 caption. The same for check-point inhibitors – ICI does not appear until line 328. The explanation for ATB should stay in line 361 (not in line 373): We have introduced the explanation.

- ‘Bacteria’ (line 191) and ‘Microbiome’ (line 193) should be lowercase: We have corrected.

- Pay attention to: ‘et al’ or ‘et al.’, placing a reference ‘… [1].’ or ‘… . [2]’ (line 68), ‘microorganisms’ or ‘micro-organisms’ (line 211), ‘checkpoint’ or ‘check-point’, Escherichia Coli (line 353). Please keep the format consistent throughout the whole text: These have been corrected

- There are some typographic errors, mainly a lack of spaces (e.g., lines 16, 69, 132 …): These have been corrected

Reviewer 2 Report

Authors present the recent evidence regarding microbiome alterations 16 in patients with gynecological tumors, to understand if the link that exists between microbiome, 17 immunity and cancer can guide the prophylactic, diagnostic and therapeutic management of gyne-18 cological cancers. Despite significant advances in understanding the pathogenetic mechanisms underlying 11 gynecological cancers, these cancers still remain widespread.

I have not found significant limitations in this proposal; conversely, I think that it has many strengths, such as originality and methodological accuracy. I think that this project I reviewed reports novel findings. Tables and figures are also fine for the paper.

Author Response

No changes were requested, thank you for the positive feedback on our work.

Reviewer 3 Report

Dear authors 

Please improve table 1 results and add references. 

please cite to these references: 

Yousefi B, Eslami M, Ghasemian A, Kokhaei P, Salek Farrokhi A, Darabi N. Probiotics importance and their immunomodulatory properties. Journal of cellular physiology. 2019 Jun;234(6):8008-18.

D’asheesh TI, Hussen BM, Al-Marzoqi AH, Ghasemian A. Assessment of oncogenic role of intestinal microbiota in colorectal cancer patients. Journal of gastrointestinal cancer. 2021 Sep;52:1016-21.

Author Response

Please improve table 1 results and add references: Completed as requested. Previous table 1 is now table 2.

please cite to these references: we have added 

Yousefi B, Eslami M, Ghasemian A, Kokhaei P, Salek Farrokhi A, Darabi N. Probiotics importance and their immunomodulatory properties. Journal of cellular physiology. 2019 Jun;234(6):8008-18.

D’asheesh TI, Hussen BM, Al-Marzoqi AH, Ghasemian A. Assessment of oncogenic role of intestinal microbiota in colorectal cancer patients. Journal of gastrointestinal cancer. 2021 Sep;52:1016-21.

Round 2

Reviewer 1 Report

The authors have adequately addressed my comments. Thank you. I recommend the publication of the paper in its present form.